# Validity of the Cancer and Aging Research Group Predictive Tool in Older Japanese Patients

**DOI:** 10.3390/cancers14092075

**Published:** 2022-04-21

**Authors:** Hirotaka Suto, Yumiko Inui, Atsuo Okamura

**Affiliations:** 1Department of Medical Oncology, The Cancer Institute Hospital of Japanese Foundation for Cancer Research, Tokyo 135-8550, Japan; 2Department of Medical Oncology/Hematology, Kakogawa Central City Hospital, Kakogawa 675-8611, Japan; yuinui@med.kobe-u.ac.jp (Y.I.); atsuo@godzilla.kobe-u.ac.jp (A.O.)

**Keywords:** Cancer and Aging Research Group predictive tool, older Japanese patients, chemotherapy-related adverse events

## Abstract

**Simple Summary:**

This study aimed to assess the usefulness of the Cancer and Aging Research Group (CARG) predictive tool in predicting chemotherapy-related adverse events (CRAEs) in elderly Japanese patients with cancer. The CARG has developed a very useful tool that can predict CRAEs among patients with solid tumors aged above 65 years. However, considering that this tool is based on data from Europe and the United States, the usefulness of the tool for Japanese people, who have different sensitivities to anticancer drugs and varying life expectancies, is still unknown. This study showed that the CARG tool could help predict CRAEs in elderly Japanese patients with cancer, i.e., this tool has the potential to optimize chemotherapy in elderly Japanese patients with cancer.

**Abstract:**

Background: This study aimed to evaluate the usefulness of the Cancer and Aging Research Group (CARG) predictive tool in older Japanese patients with cancer. Methods: Patients aged 65 years or older with solid tumors treated with new anticancer regimens in Kakogawa Central City Hospital between April 2016 and March 2019 were included. Grade 3 or higher risks of developing chemotherapy-related adverse events (CRAEs) were calculated using the tool (low-, intermediate-, or high-risk scores). The association between grade 3–5 CRAE incidence during the first course of each regimen and the calculated risk or the patient characteristics was evaluated. The difference in the incidences of CRAEs between the groups was evaluated by Fisher’s exact test. Results: This study examined 76 patients (mean age: 71 (65–82) years). The incidence of grade 3–5 CRAE was 38%, 55%, and 76% in patients classified as low, medium, and high CARG risk scores (*p* = 0.035), and the incidence of severe non-hematological toxicities was 4%, 31%, and 52% (*p* < 0.01), respectively. Eastern Cooperative Oncology Group performance status and age were not associated with chemotherapy toxicity. Conclusions: The CARG predictive tool was valid, suggesting its usefulness in optimizing chemotherapy outcomes in older patients with cancer.

## 1. Introduction

In Japan, patients with cancer aged 65 and over account for approximately 70% of all cancer cases [1]. Consequently, anticancer drug treatment is increasingly needed for older individuals. Older patients with cancer have a similar efficacy of anticancer drug therapy to non-older patients [2,3,4]; however, they tend to have a higher incidence and severity of chemotherapy-related adverse events (CRAEs) [5,6], owing to age-related decline in organ and body functions, comorbidities, and polypharmacy [7,8,9,10]. Therefore, unlike in non-older patients with cancer, using age and Eastern Cooperative Oncology Group performance status (ECOG PS) alone in older counterparts to determine the indication for anticancer therapy is inadequate.

The Cancer and Aging Research Group (CARG) has developed a CRAE prediction tool that is very useful for patients aged over 65 years with solid tumors [11,12]. This tool can predict the occurrence of grade 3–5 CRAEs in older patients with solid tumors, which are difficult to predict by age and ECOG PS. Recently, the American Society of Clinical Oncology guideline for geriatric oncology has recommended this tool for predicting the side effects of anticancer drug therapy in older patients with cancer [13].

However, given that this tool is based on data from Europe and the United States, its usefulness for Japanese people, who have different sensitivities to anticancer drugs and life expectancies, is still unknown [14,15]. Hence, this study aimed to evaluate the usefulness of the CARG predictive tool in older Japanese patients with cancer.

## 2. Materials and Methods

### 2.1. Study Population

Patients aged 65 years or older with solid tumors who received a new anticancer drug regimen in the Department of Medical Oncology/Hematology at Kakogawa Central City Hospital between April 2016 and March 2019 were included. Conversely, those who received concurrent radiation or treatment for a clinical trial were excluded.

The study was approved by the Institutional Review Board Kakogawa Central City Hospital Ethics Committee (no. 2019079). The study conformed to the principles of the Declaration of Helsinki of 1964 and later versions. Given the retrospective nature of this study, our hospital’s institutional review board waived the requirement for the patients’ informed consent.

### 2.2. Study Design

Before chemotherapy, patients completed a medical questionnaire used in daily clinical practice. The contents of the questionnaire are as follows: comorbidities, hearing impairment, falls in the past 6 months, walking restriction for 100 m, need for medication assistance, and social activity loss caused by physical and mental health. We also recorded the patients’ tumor characteristics (tumor type and stage), pretreatment laboratory data (complete blood count, creatinine, and liver function tests), chemotherapy regimen, line of chemotherapy (first line or later), use of granulocyte colony-stimulating factor (G-CSF), and drug and dosing of the first chemotherapy treatment. As per the American National Comprehensive Cancer Network guidelines, the chemotherapy dosing for the first treatment cycle was categorized as standard or dose reduced. Grade 3–5 CRAEs during chemotherapy were defined according to the National Cancer Institute Common Terminology Criteria for Adverse Events, version 5.0, and were determined through medical record review. Laboratory-based toxicities were identified according to laboratory values obtained on the date of scheduled chemotherapy or during the time when the patient sought medical care for symptoms between chemotherapy cycles.

### 2.3. Statistical Analysis

For each patient, a chemotherapy toxicity score was calculated using the 11 prechemotherapy variables included in the CARG predictive tool for chemotherapy toxicity (Appendix A) [11,12]. Patients were categorized as having a low (0–5 points), moderate (6–9 points), or high (≥10 points) toxicity risk based on specified cut-points [11,12]. Subsequently, we evaluated the association between the incidence of grade 3 or higher CRAEs during the first course of each regimen and the CARG risk score or the patient characteristics. This distribution of toxicity over the different risk groups was compared with the ability of ECOG PS or age to predict toxicity. Patients were classified according to three ECOG PS scores (0, 1, and ≥2) and three age groups (65–69, 70–74, and ≥75 years).

The difference in the incidences of grade 3–5 toxicity between groups was evaluated by Fisher’s exact test. All statistical data were analyzed using EZR (Saitama Medical Center, Jichi Medical University, Saitama, Japan) [16]. A *p* value of less than 0.05 was considered statistically significant.

## 3. Results

### 3.1. Patient Characteristics

This study included 76 consecutive solid tumor patients aged 65 years or older who received a new anticancer drug regimen at the department (Table 1), with a median age of 71 (65–82) years and a male predominance of 58%. The most common cancer type was gastrointestinal tumors (54%). In addition, 47% of all patients received polychemotherapy, and 70% received standard doses of chemotherapy. A majority of the patients had an ECOG PS of 1 (59%).

### 3.2. CRAEs

The most common grade 3–5 hematologic toxicities were neutropenia (28%), leucopenia (18%), and anemia (14%), whereas the most common grade 3–5 non-hematologic toxicities were fatigue (16%), nausea (16%), and mucositis oral (4%) (Appendix A). The most common grade 3–5 hematologic toxicities by CARG risk score were as follows: neutropenia (35%), neutropenia (31%), and anemia (43%) in the low-, intermediate-, and high-risk groups, respectively. The most common grade 3–5 non-hematologic toxicities were fatigue (4%) in the low-risk group, nausea (7%) in the intermediate-risk group, and fatigue (48%) and nausea (48%) in the high-risk group, according to CARG risk scores (Table 2). There was no chemotherapy-induced death.

### 3.3. Ability of CARG Risk Score vs. ECOG PS vs. Age to Predict Grade 3–5 CRAEs

The incidence of grade 3–5 CRAEs in patients classified as having a low, medium, or high CARG risk score was 38%, 55%, and 76%, respectively (*p* = 0.035) (Figure 1a). When classified according to the patients’ ECOG PS, the incidence was 53%, 49%, and 75% in those with a PS of 0, 1, and 2 or higher, respectively (*p* = 0.33) (Figure 1b). By age, the incidence was 67%, 38%, and 52% for those who were 65–69, 70–74, and 75 years old and above, respectively, but no association was found between age and incidence (*p* = 0.22) (Figure 1c). There was also no association between CCI and the incidence of grade 3–5 CRAEs (Appendix A).

The incidence of severe hematological toxicity was 35%, 31%, and 57% (*p* = 0.15) (Figure 2a) and that of severe non-hematological toxicities was 4%, 31%, and 52% (*p* < 0.01) (Figure 2b) in patients with low, medium, and high CARG risk scores, respectively. Alternatively, there was no association between ECOS PS or age and the incidence of severe hematological or non-hematological toxicity (Appendix A).

## 4. Discussion

Approximately 70% of Japanese patients with cancer were 65 years old and above [1]. However, most of the evidence used in treating older patients with solid tumors is extrapolated from the data of younger patients because of the lack of good-quality evidence specific to the older population [17,18,19]. Older patients with cancer still have no universal index for predicting CRAEs. In the United States and Europe, age has no association with ECOG PS in predicting CRAEs in older patients with cancer, suggesting that the CARG predictive tool is superior in predicting CRAEs [11,12].

Likewise, our study suggested that the CARG predictive tool was superior in predicting grade 3–5 CRAEs even in older Japanese patients with cancer. Similar to overseas reports, predicting severe CRAEs by age and ECOG PS is difficult, probably because these classifications alone do not adequately assess the decline and vulnerability of organs and body functions in people over 65 years old. As for CCI, the incidence of adverse events did not appear to correlate with CCI because most subjects in this study were assigned to the very high-risk group.

Recently, several studies have reported on the CARG predictive tool specific to various countries and cancer types [20,21,22,23,24,25,26,27,28]. Some studies reported that the CARG predictive tools helped predict severe CRAEs [20,21,22,23,24,25], while others reported that the CARG predictive tool did not help predict severe CRAEs [26,27,28]. Moth et al. reported no association between the risk assessment by the CARG predictive tool and severe CRAEs [26]. However, in their study, prophylactic G-CSF was not administered, which may have resulted in an overestimation of hematologic toxicity. The skewed intermediate-risk rate of 61% may have also influenced the results. The study by Chan et al. also reported no association between the risk assessment by the CARG predictive tool and severe CRAE [27]. In their study, the percentages of low-, intermediate-, and high-risk groups by the CARG risk score were 33.6%, 37.1%, and 29.3%, respectively, similar to the population in the Hurria et al. study [11]. However, in the study by Chan et al., approximately 20% of patients did not receive chemotherapy, and it was unclear which risk group they belonged to, which may have compromised the usefulness of this tool. The study by Alibhai et al. was specific to prostate cancer patients, with approximately 95% of patients receiving docetaxel monotherapy [28]. Furthermore, severe CRAE occurred in only nine patients, making it challenging to evaluate this tool to predict toxicity.

Key factors that contributed to demonstrating the usefulness of the CARG tool in this study include the following. First, the patient characteristics are comparable to those in the research by Hurria et al. [11]. Namely, the patient characteristics are similar to those of Hurria et al. in terms of the proportion of low-risk, intermediate-risk, and high-risk groups according to the CARG risk score and the percentage of supportive care, such as prophylactic G-CSF administration. If the proportions of each group are excessively different, the frequency of both hematologic and non-hematologic toxicities may differ significantly from the predictions. Second, we limited the assessment of severe toxicity to the first cycle of chemotherapy only. Some chemotherapy toxicity is cumulative [29,30]. Therefore, the incidence of severe toxicity increases with the course of treatment. Furthermore, in many clinical trials, toxicity tends to occur more frequently in Asians than in Westerners [31,32,33,34,35,36,37]. In the study by Yoshida et al., in older Japanese patients with gynecologic cancer, the cumulative incidence of grade 3 or higher adverse events was 58% in the low-risk group, 80% in the intermediate-risk group, and 100% in the high-risk group of the CARG risk score, indicating a very high incidence of severe toxicity in all risk groups [24]. Therefore, it seemed unlikely that there would be group differences in cumulative toxicity rates in the older Japanese population.

Other functional assessment tools for older patients with cancer include G8 and VES-13 screening tools [38,39], but they have not been developed to predict CRAEs. Thus, previous studies have reported that G8 and VES-13 have not been able to predict severe CRAEs [21,27,28]. Another tool for predicting CRAEs is the CRASH score [40], but it cannot evaluate the toxicity of new therapies. As for the CARG predictive tool, it can be used for new cytotoxic anticancer therapies.

Importantly, the current study showed that the CARG risk score was associated more strongly with the incidence of severe non-hematologic toxicity than that of severe hematologic toxicity. In other previous studies using the CARG predictive tool, few data have been published on whether there was an association between hematologic toxicity and CARG risk score [20,21,22,23,24,25,26,27,28]. Ortland et al. also reported no association between the CARG risk score and hematologic toxicity [23]. However, we should note that in their study, approximately 40% of the patients had hematologic malignancies, which is a different patient population and treatment modality. Among the study population, a standard anticancer drug dosage rate of approximately 70%, monotherapy rate of approximately 50%, and G-CSF usage rate of approximately 15% were discovered, which might explain the observed reduced incidence of severe hematologic toxicity. In addition, our study evaluated CRAE only in the first cycle and not cumulative myelosuppression. Therefore, severe hematologic toxicity might have been underestimated and was not associated with the CARG risk score.

This study has some limitations. First, it is a retrospective study with a relatively small sample size. Therefore, it is not possible to compare the usefulness of the CARG predictive tool with other tools, such as the CRASH score, if the questionnaire does not include the numerous items required for the Mini Mental State Examination or the Mini Nutritional Assessment. A multicenter prospective study is needed to validate the usefulness of the CARG predictive tool. Second, accumulative and delayed toxicity was insufficiently evaluated. Therefore, a longer observation period is needed for dose-dependent and delayed toxicity. Third, it is currently unknown whether new molecular targeted drugs or immune checkpoint inhibitors can predict toxicity. Therefore, it is necessary to accumulate data from elderly cancer patients receiving new molecular targeted drugs and immune checkpoint inhibitors to validate the usefulness of the CARG predictive tool. Lastly, an insight into the extent to which anticancer drug doses should be reduced in groups classified as high risk by the CARG predictive tool remains undetermined. Different cancer types require different methods of reducing the anticancer drug dose. For example, whether to use multidrug therapy or monotherapy of anticancer drugs or molecular targeted drugs for older patients with colorectal cancer remains controversial [41,42]. Nonetheless, the GAP70+ trial reported that anticancer drug reduction based on geriatric assessment for older patients with cancer mitigated CRAEs and did not significantly decrease survival [43]; thus, it might be helpful to use the geriatric assessment tool. Specifically, in the current study, this tool predicted the toxicity of the first treatment cycle; thus, it can be used as an indicator to reduce the dose of anticancer drugs before starting the treatment.

## 5. Conclusions

The CARG predictive tool appears to be valid in older Japanese patients with cancer. In particular, this tool may facilitate the prediction of non-hematological toxicity. If the tool predicts a high incidence of severe CRAEs, reducing the dosage of anticancer drugs in older patients with cancer might be necessary.

## Figures and Tables

**Figure 1 cancers-14-02075-f001:**
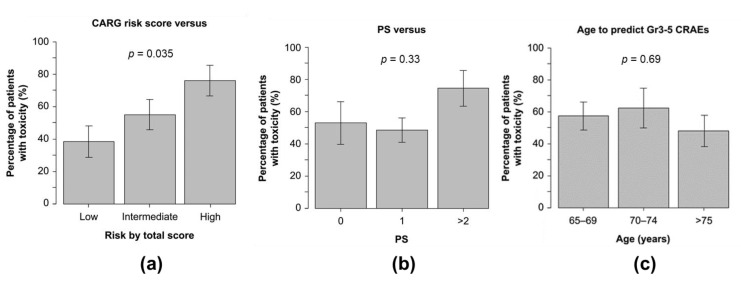
Ability of (**a**) CARG risk score vs. (**b**) ECOG PS vs. (**c**) age to predict grade 3–5 CRAEs. Abbreviations: CARG, Cancer and Aging Research Group; CRAEs, chemotherapy-related adverse events; ECOG PS, Eastern Cooperative Oncology Group performance status.

**Figure 2 cancers-14-02075-f002:**
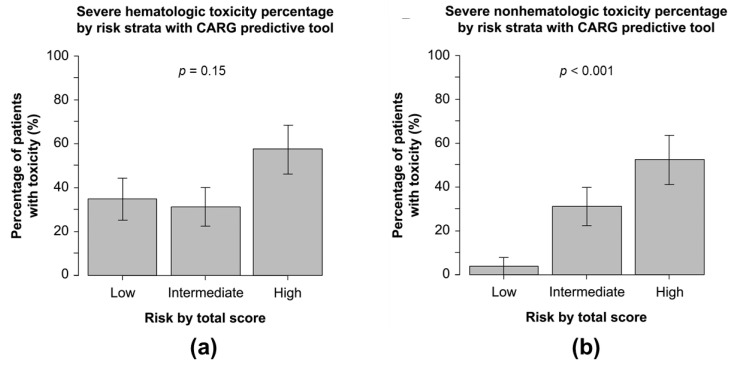
The percentage of participants with severe hematologic (**a**) and non-hematologic (**b**) toxicity based on CARG predictive tool risk strata. Abbreviations: CARG, Cancer and Aging Research Group.

**Table 1 cancers-14-02075-t001:** Patient characteristics.

Characteristics	No.	%
Age (years)		
65–69	33	43
70–74	16	21
75–79	20	26
≥80	7	9
Sex		
Male	44	58
Female	32	42
ECOG PS		
0	15	20
1	45	59
≥2	16	21
Cancer type		
Breast	7	9
Lung	6	8
GI	41	54
GYN	3	4
GU	1	1
Other	18	24
Stage		
I	2	3
II	2	3
III	8	10
IV	63	83
Other	1	1
Treatment		
Standard dose		
Yes	53	70
No	23	30
No. of chemo drugs		
Monochemotherapy	40	53
Polychemotherapy	36	47
Line of chemotherapy		
First line	25	33
≥Second line	51	67
Growth factor use		
Yes	11	14
No	65	86
Hemoglobin		
<10 g/dL (female)	16	21
≥10 g/dL (female)	16	21
<11 g/dL (male)	20	26
≥11 g/dL (male)	24	32
Creatinine clearance		
<34 mL/min	5	7
≥34 mL/min	71	93
Hearing		
Fair, poor, or totally deaf	4	5
Excellent or good	72	95
No. of falls in the past 6 months		
≥1	1	1
None	75	99
Taking medications		
With some help/unable	10	13
Without help	66	87
Limited in walking 100 m		
Somewhat limited/limited a lot	24	32
Not limited	52	68
Decreased social activity because of		
health/emotional problems		
Some, most, all of the time	27	36
A little, or none of the time	49	64
CCI		
0 (low)	0	0
1–2 (medium)	4	5
3–4 (high)	7	9
≥5 (very high)	65	86
CARG		
0–5 (low)	26	34
6–9 (intermediate)	29	38
≥10 (high)	21	28

CARG, Cancer and Aging Research Group; CCI, Charlson comorbidity index; ECOG PS, Eastern Cooperative Oncology Group performance status; GYN, gynecologic.

**Table 2 cancers-14-02075-t002:** Chemotherapy-related adverse events by CARG risk score.

Grade 3–5 CRAEs	CARG Risk Score Low (0–5) No. %	CARG Risk Score Intermediate (6–9) No. %	CARG Risk Score High (≥10) No. %
Hematologic			
Leukopenia	6 23	4 14	4 19
Neutropenia	9 35	9 31	3 14
Anemia	1 4	1 3	9 43
Thrombocytopenia	1 4	0 0	1 5
Febrile neutropenia	2 8	0 0	1 5
Non-hematologic			
Fatigue	1 4	1 3	10 48
Nausea	0 0	2 7	10 48
Mucositis oral	0 0	0 0	3 14
Diarrhea	0 0	1 3	0 0
Hypertension	0 0	1 3	0 0
Proteinuria	0 0	1 3	0 0
Edema	0 0	0 0	1 5
Hyponatremia	0 0	1 3	1 5
Hyperkalemia	0 0	1 3	1 5
Hypomagnesemia	0 0	1 3	0 0

CRAEs, chemotherapy-related adverse events; CARG, Cancer and Aging Research Group.

## Data Availability

All data generated or analyzed during this study are included in the published article.

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
