# Peer review of "Validity of the Cancer and Aging Research Group Predictive Tool in Older Japanese Patients"

_cancers, 2022, doi:10.3390/cancers14092075_

Round 1
Reviewer 1 Report
Accept as is
Author Response
Response to Reviewer 1
Reviewer 1:
Accept as is.
Response:
Thank you for this positive feedback.
Reviewer 2 Report
Validity of the Cancer and Aging Research Group Predictive Tool in Older Japanese Patients
I have several major comments
The inclusion period is very long (36 months) for only 76 patients ie 2 elderly patients/months. Can the authors report if all the consecutive patients were included or if there is some selection?
In the end of Study population paragraph, authors stated that the study is retrospective but in the following paragraph, it’s appears that assessment was done before chemotherapy, can the authors clarify?
In the statistical analysis, authors used the term correlation, I think the term association will be more relevant
In the abstract and also in the discussion, authors should highlight that there is a significant association for no hematologic but not for hematologic AE. This result is concordant with others studies?
Minor comment
- Abstract: method: tests used for the correlation should be reported
- The table 1 is not original data’s, and can be move as supplementary data’s
- At the end of paragraph 3.2 I suggest to rewrite the sentence as “there is no chemotherapy induced death”
- Table 3 is not really informative, I suggest to report the grade 3-5 AE according to the 3 categories of CRAE
- There is some typo problem with figure 1, it’s appears after figure 2 and the titles appears as incomplete
Author Response
Response to Reviewer 2
Reviewer 2:
I have several major comments
Response:
Thank you for your careful and very constructive review.
The inclusion period is very long (36 months) for only 76 patients ie 2 elderly patients/months. Can the authors report if all the consecutive patients were included or if there is some selection?
Response:
The study included 76 consecutive patients over the age of 65 with solid tumors treated in our department. Therefore we have changed the following text in the Results section.
“This study included 76 consecutive solid tumor patients aged 65 years or older who received a new anticancer drug regimen at the department (Table 1), with a median age of 71 (65–82) years and a male predominance of 58%.” (lines 99-101)
In the end of Study population paragraph, authors stated that the study is retrospective but in the following paragraph, it’s appears that assessment was done before chemotherapy, can the authors clarify?
Response:
This is a retrospective study. It utilizes a pretreatment questionnaire used in routine clinical practice. We have added the following statement in the study design section.
“Before chemotherapy, patients completed a medical questionnaire used in daily clinical practice. The contents of the questionnaire are as follows.” (lines 67-68)
In the statistical analysis, authors used the term correlation, I think the term association will be more relevant.
Response:
We have changed "correlation" to "association" in the paper.
In the abstract and also in the discussion, authors should highlight that there is a significant association for no hematologic but not for hematologic AE. This result is concordant with others studies?
Response:
We have added the following sentences to emphasize that we found an association between severe non-hematologic toxicities and CARG risk scores.
“the incidence of severe non-hematological toxicities was 4%, 31%, and 52% (p < 0.01), respectively.” (lines 28-29)
“In particular, this tool may facilitate the prediction of non-hematological toxicity.” (lines 235-236)
In addition, many other studies did not report data separating non-hematologic and hematologic toxicity. Ortland et al. reported data separating non-hematologic and hematologic toxicity, but also reported no association between CARG risk score and hematologic toxicity. Therefore, we have added the following text to the discussion section. “In other previous studies using the CARG predictive tool, few data have been published on whether there was an association between hematologic toxicity and CARG risk score. Ortland et al. also reported no association between CARG risk score and hematologic toxicity. However, we should note that in their study, approximately 40% of the patients had hematologic malignancies, which is a different patient population and treatment modality.” (lines 200-205)
Minor comment
Response:
Thank you for your kind comments.
1.Abstract: method: tests used for the correlation should be reported
Response:
We have added a description of the Fisher test in the Methods section of the abstract.
- The table 1 is not original data’s, and can be move as supplementary data’s
Response:
We have moved Table 1 to the Supplemental Data as Table S1.
- At the end of paragraph 3.2 I suggest to rewrite the sentence as “there is no chemotherapy induced death”
Response:
We have rewritten the sentence at the end of section 3.2 to read "There is no chemotherapy induced death.".
- Table 3 is not really informative, I suggest to report the grade 3-5 AE according to the 3 categories of CRAE
Response:
We have modified Table 3 to grade 3-5 AEs according to the CARG risk scores.
- There is some typo problem with figure 1, it’s appears after figure 2 and the titles appears as incomplete
Response:
We have corrected the position of Figure 1 in our paper.
Thank you again for taking the time to review the manuscript.
Reviewer 3 Report
In this retrospective study, the authors evaluated retrospectively the impact of the CARG score, previously developed in 2011 and validated in 2016 by Arti Hurria, in a Japanese population.
In a rapid screening on pubmed, numerous articles have performed equivalent studies in numerous "country-specific" populations:
- 2013: Predicting chemotherapy toxicity in older adults with lung cancer.
J Geriatr Oncol. 2013 Oct;4(4):334-9. doi: 10.1016/j.jgo.2013.05.002. Epub 2013 Jun 14; PMID: 24472476- 2017: A comparison of the CARG tool, the VES-13, and oncologist judgment in predicting grade 3+ toxicities in men undergoing chemotherapy for metastatic prostate cancer.
J Geriatr Oncol. 2017 Jan;8(1):31-36. doi: 10.1016/j.jgo.2016.09.005. Epub 2016 Oct 15; PMID: 27756545 - 2019: Predicting chemotherapy toxicity in older adults: Comparing the predictive value of the CARG Toxicity Score with oncologists' estimates of toxicity based on clinical judgement. J Geriatr Oncol. 2019 Mar;10(2):202-209. doi: 10.1016/j.jgo.2018.08.010. Epub 2018 Sep 14; PMID: 30224184 - 2019: The performance of three oncogeriatric screening tools - G8, optimised G8 and CARG - in predicting chemotherapy-related toxicity in older patients with cancer. A prospective clinical study. J Geriatr Oncol. 2019 Nov;10(6):937-943. doi: 10.1016/j.jgo.2019.04.004. Epub 2019 May 10; PMID: 31085136 - 2019: Prospective comparison of the value of CRASH and CARG toxicity scores in predicting chemotherapy toxicity in geriatric oncology. Oncol Lett. 2019 Nov;18(5):4947-4955. doi: 10.3892/ol.2019.10840. Epub 2019 Sep 10; PMID: 31612006 - 2020: Comparing the performance of the CARG and the CRASH score for predicting toxicity in older patients with cancer. J Geriatr Oncol. 2020 Jul;11(6):997-1005. doi: 10.1016/j.jgo.2019.12.016. Epub 2020 Jan 9; PMID: 31928942 - 2020: Predicting toxicity of platinum and taxane-based chemotherapy in older patients with gynecologic cancer. J BUON. 2020 Mar-Apr;25(2):736-742. -2021: The predictive value of G8 and the Cancer and aging research group chemotherapy toxicity tool in treatment-related toxicity in older Chinese patients with cancer. J Geriatr Oncol. 2021 May;12(4):557-562. doi: 10.1016/j.jgo.2020.10.013. Epub 2020 Oct 27; PMID: 33127385 - 2021: Cancer Aging Research Group (CARG) score in older adults undergoing curative intent chemotherapy: a prospective cohort study. BMJ Open. 2021 Jun 29;11(6):e047376. doi: 10.1136/bmjopen-2020-047376.PMID: 34187825 Most of these reports did not only evaluate CARG score performances but compared it to those of the CRASH score or other geriatric tools. In their current form, the results are not original and an analysis of the results obtained to this abundant literature about CARG score performances should at least be added to the discussion of the paper. For example, it is surprising that the differences between the 3 risk groups are attenuated compared to the development cohort, and close to those of the validation cohort of Hurria et al. This point should be analysed and hypotheses should be proposed to explain the differences observed in the various popuulations tested in the literature: does the ethnicity of the patients explain part of these differences? or the type of treatment dispensed? or the geriatric characteristics of the population?Author Response
Response to Reviewer 3
Reviewer 3:
In this retrospective study, the authors evaluated retrospectively the impact of the CARG score, previously developed in 2011 and validated in 2016 by Arti Hurria, in a Japanese population.
In a rapid screening on pubmed, numerous articles have performed equivalent studies in numerous "country-specific" populations:
- 2013: Predicting chemotherapy toxicity in older adults with lung cancer.
Nie X, Liu D, Li Q, Bai C.J Geriatr Oncol. 2013 Oct;4(4):334-9. doi: 10.1016/j.jgo.2013.05.002. Epub 2013 Jun 14; PMID: 24472476
- 2017: A comparison of the CARG tool, the VES-13, and oncologist judgment in predicting grade 3+ toxicities in men undergoing chemotherapy for metastatic prostate cancer.
Alibhai SM, Aziz S, Manokumar T, Timilshina N, Breunis H.J Geriatr Oncol. 2017 Jan;8(1):31-36. doi: 10.1016/j.jgo.2016.09.005. Epub 2016 Oct 15; PMID: 27756545 - 2019: Predicting chemotherapy toxicity in older adults: Comparing the predictive value of the CARG Toxicity Score with oncologists' estimates of toxicity based on clinical judgement. Moth EB, Kiely BE, Stefanic N, Naganathan V, Martin A, Grimison P, Stockler MR, Beale P, Blinman P.J Geriatr Oncol. 2019 Mar;10(2):202-209. doi: 10.1016/j.jgo.2018.08.010. Epub 2018 Sep 14; PMID: 30224184 - 2019: The performance of three oncogeriatric screening tools - G8, optimised G8 and CARG - in predicting chemotherapy-related toxicity in older patients with cancer. A prospective clinical study. Kotzerke D, Moritz F, Mantovani L, Hambsch P, Hering K, Kuhnt T, Yahiaoui-Doktor M, Forstmeyer D, Lordick F, Knödler M.J Geriatr Oncol. 2019 Nov;10(6):937-943. doi: 10.1016/j.jgo.2019.04.004. Epub 2019 May 10; PMID: 31085136 - 2019: Prospective comparison of the value of CRASH and CARG toxicity scores in predicting chemotherapy toxicity in geriatric oncology. Zhang J, Liao X, Feng J, Yin T, Liang Y.Oncol Lett. 2019 Nov;18(5):4947-4955. doi: 10.3892/ol.2019.10840. Epub 2019 Sep 10; PMID: 31612006 - 2020: Comparing the performance of the CARG and the CRASH score for predicting toxicity in older patients with cancer. Ortland I, Mendel Ott M, Kowar M, Sippel C, Jaehde U, Jacobs AH, Ko YD.J Geriatr Oncol. 2020 Jul;11(6):997-1005. doi: 10.1016/j.jgo.2019.12.016. Epub 2020 Jan 9; PMID: 31928942 - 2020: Predicting toxicity of platinum and taxane-based chemotherapy in older patients with gynecologic cancer. Yoshida H, Shintani D, Kawashima N, Fujiwara K.J BUON. 2020 Mar-Apr;25(2):736-742. -2021: The predictive value of G8 and the Cancer and aging research group chemotherapy toxicity tool in treatment-related toxicity in older Chinese patients with cancer. Chan WL, Ma T, Cheung KL, Choi H, Wong J, Lam KO, Yuen KK, Luk MY, Kwong D.J Geriatr Oncol. 2021 May;12(4):557-562. doi: 10.1016/j.jgo.2020.10.013. Epub 2020 Oct 27; PMID: 33127385 - 2021: Cancer Aging Research Group (CARG) score in older adults undergoing curative intent chemotherapy: a prospective cohort study. Ostwal V, Ramaswamy A, Bhargava P, Hatkhambkar T, Swami R, Rastogi S, Mandavkar S, Ghosh J, Bajpai J, Gulia S, Srinivas S, Rath S, Gupta S.BMJ Open. 2021 Jun 29;11(6):e047376. doi: 10.1136/bmjopen-2020-047376.PMID: 34187825 Most of these reports did not only evaluate CARG score performances but compared it to those of the CRASH score or other geriatric tools. In their current form, the results are not original and an analysis of the results obtained to this abundant literature about CARG score performances should at least be added to the discussion of the paper. For example, it is surprising that the differences between the 3 risk groups are attenuated compared to the development cohort, and close to those of the validation cohort of Hurria et al. This point should be analysed and hypotheses should be proposed to explain the differences observed in the various popuulations tested in the literature: does the ethnicity of the patients explain part of these differences? or the type of treatment dispensed? or the geriatric characteristics of the population?
Response:
Thank you for your careful and very constructive review. We have added the studies you mentioned to our discussion and added the following explanation as to why some studies had poor CARG score results and the reason why the results of the current study were similar to Hurria's data.
“Recently, several studies have reported on the CARG predictive tool specific to various countries and cancer types. Some studies reported that the CARG predictive tools helped predict severe CRAEs, while others reported that the CARG predictive tool did not help predict severe CRAEs. Moth et al. reported no association between risk assessment by the CARG predictive tool and severe CRAEs. However, in their study, prophylactic G-CSF was not administered, which may have resulted in an overestimation of hematologic toxicity. The skewed intermediate-risk rate of 61% may also have influenced the results. The study by Chan et al. also reported no association between risk assessment by the CARG predictive tool and severe CRAE. In their study, the percentages of low, intermediate, and high-risk groups by CARG risk score were 33.6%, 37.1%, and 29.3%, respectively, similar to the population in the Hurria et al. study. However, in the study by Chan et al., approximately 20% of patients did not receive chemotherapy, and it was unclear which risk group they belonged to, which may have compromised the usefulness of this tool. The study by Alibhai et al. was specific to prostate cancer patients, with approximately 95% of patients receiving docetaxel monotherapy. Furthermore, severe CRAE occurred in only nine patients, making it challenging to evaluate this tool to predict toxicity.
Key factors that contributed to demonstrating the usefulness of the CARG tool in this study include the following. First, the patient characteristics are comparable to those in the research by Hurria et al.. Namely, the patient characteristics are similar to that of Hurria et al. in terms of the proportion of low-risk, intermediate-risk, and high-risk groups according to the CARG risk score and the percentage of supportive care such as prophylactic G-CSF administration. If the proportions of each group are excessively different, the frequency of both hematologic and non-hematologic toxicities may differ significantly from the predictions. Second, we limited the assessment of severe toxicity to the first cycle of chemotherapy only. Some chemotherapy toxicity is cumulative. Therefore, the incidence of severe toxicity increases with the course of treatment. Furthermore, in many clinical trials, toxicity tends to occur more frequently in Asians than in Westerners. In the study by Yoshida et al. in older Japanese patients with gynecologic cancer, the cumulative incidence of grade 3 or higher adverse events was 58% in the low-risk group, 80% in the intermediate-risk group, and 100% in the high-risk group of the CARG risk score, indicating a very high incidence of severe toxicity in all risk groups. Therefore, it seemed unlikely that there would be group differences in cumulative toxicity rates in the older Japanese population.” (lines 158-191)
We have noted the pros and cons of the CRASH score and other geriatrics tools. We have added the following text to the Discussion section in the revised manuscript.
“Other functional assessment tools for older patients with cancer include G8 and VES-13 screening tools, but they have not been developed to predict CRAEs. Thus, previous studies have reported that G8 and VES-13 have not been able to predict severe CRAEs.” (lines 192-195)
Thank you again for taking the time to review th
Round 2
Reviewer 2 Report
Thanks for yours reply